# MEOW: - Automatic Evolutionary Multi-Objective Concealed Weapon Detection

**Daniel Dimanov**[1]  **Colin Singleton**[2]  **Shahin Rostami**[3]  **Emili Balaguer-Ballester**[1]

[1]Department of Computing and Informatics, Bournemouth University, UK
[2]CountingLab Limited, University of Reading, UK
[3]Data Science Lab, Polyra Limited, Bournemouth, UK

**Abstract**  X-ray screening is crucial for ensuring safety and security in crowded public areas. However, X-ray operators are often overwhelmed by the sheer amount of potential threats to assess; thus, current computer vision-aided systems are designed to alleviate these workloads. In this study, we focus on a key, unresolved challenge for developing such automatic X-ray screening systems: the direct application of existing *avant garde* computer vision approaches does not necessarily yield satisfactory results in the X-ray medium, hindering the effectiveness of current screening systems. To overcome this drawback, we propose a novel automated machine learning (AutoML) multi-objective approach for neural architecture search (NAS) for concealed weapon detection (MEOW). We benchmark MEOW with the state-of-the-art in two comprehensive scenarios in threat identification: *SIXray* (a popular, massive X-ray dataset) and *Residuals* (a proprietary, unpublished dataset provided by our industry partners). MEOW consist of the coalescence of two new components: First, we design a heuristic technique to strongly reduce the high computational cost of neuroevolutionary search while preserving a high performance such that it can be effectively used in real-time industrial settings. Second, we devise a novel ensemble approach for combining multiple discovered architectures simultaneously. Leveraging these two characteristics, MEOW outperforms the state-of-the-art while keeping the NAS overhead to a minimum. More broadly, our results suggest that AutoML has a strong potential for security applications.

## 1 Introduction

Concealed weapon detection is a crucial component of public security in crowded areas (Mahajan and Padha, 2018), enabling authorities to address global issues such as terrorism (Sheen et al., 2001), school shootings (Freilich et al., 2022) and weapons trafficking (Langlois et al., 2022).

Most state-of-art algorithms in this domain focus on identifying only the presence of a threat -which is a sub-optimal approach since distinct threats require different security protocols (Morris et al., 2018; Petrozziello and Jordanov, 2019). Recently, some new algorithms have addressed this problem by combining deep learning with evolutionary computation to form a promising approach to identify different threats with sufficiently high accuracy (Rostami et al., 2015).

X-Ray image input is typically employed to detect, identify and locate such threats (Miao et al., 2019). Critically, since this medium differs greatly from conventional images, the naive application of proven computer vision algorithms often yields sub-optimal results (Mery and Arteta, 2017). Hence, domain-specific techniques are necessary (Miao et al., 2019; Mery et al., 2015). However, developing new deep architectures and algorithms is typically a complex process, often requiring adaptive tuning and data-specific operations involving costly computational and human resources.

This challenge motivated the recent development of techniques that promise to be invariant to changes in data distributions and other external settings (He et al., 2021; Karmaker et al., 2021). The idea behind such approaches is to harness the computational power available nowadays to support and even partially replace a multidisciplinary team of domain experts tackling a problem

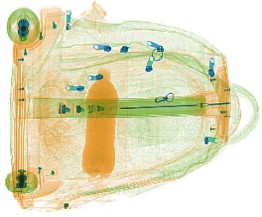 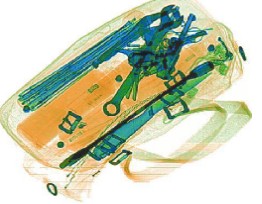 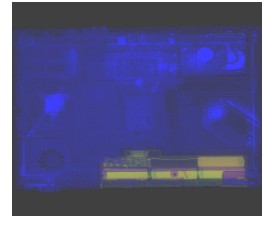 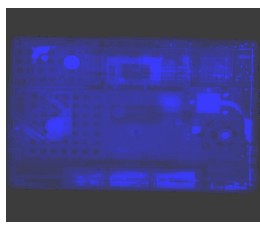

| (a) SIXray sample with no threats | (b) SIXray sample with multiple threats | (c) Residuals sample with a threat | (d) Residuals sample without threats |

Figure 1: Sample images from the two datasets.

manually. Automated machine learning (AutoML) allows this to be possible. The progress in this field led to some remarkable breakthroughs, e.g., Real et al. (2017); Lu et al. (2019); Zhou et al. (2020). Nevertheless, a significant caveat of AutoML algorithms is their extreme computational requirements (Lu et al., 2019; Real et al., 2017).

A potential approach to deal with this high computational demand is to incorporate optimisation heuristics in the form of proxy scores (Mellor et al., 2021; Abdelfattah et al., 2021).

However, security settings require not only high-performing models but also compatible and fast ones that can operate on a wide range of systems and provide predictions in real-time (Tawalbeh et al., 2020). Therefore, it is fundamental to consider multiple objectives simultaneously in the optimization process of neural network architectures for concealed weapon detection. While single-objective optimization methods can discover high-performing models, they are typically suboptimal under real-world constraints. Specifically, in this setting, it is often preferable to sacrifice some performance for efficiency(Xiong et al., 2020). To this end, multi-objective optimization (MOO) is a valuable tool for discovering neural network architectures that can achieve an optimal balance between multiple objectives, such as accuracy and overall efficiency(Dimanov et al., 2022; Lu et al., 2019).

Here, we designed a new automated multi-objective approach for neural architecture search (NAS) in concealed weapon detection problems (MEOW) and benchmarked it with two large X-ray datasets that feature different threats. The main contributions of MEOW are the following:

1. A multi-objective optimisation of several proxy scores simultaneously, rendering up to a 200x speed up w.r.t. standard multi-objective AutoML algorithms (Lu et al., 2019; Shaw et al., 2019).

2. A novel ensemble approach, utilising individual predictions from several optimal models discovered by a multi-objective NAS to provide a final, more accurate prediction.

3. The coalescence of these two components in MEOW outperforms the state-of-the-art in the well-known SIXray dataset and in a proprietary security-critical dataset in weapon detection.

## 2 Related work

Concealed weapon detection is typically performed using millimetre wave (Goenka and Sitara, 2022; Li and Wu, 2022) , radar (Zhang, 2022) and, most commonly, X-ray machines (Akcay et al., 2018; Nguyen et al., 2022). X-ray images are colour-coded based on objects' densities, facilitating visual inspection (Abidi et al., 2006; Sidky et al., 2011). Thus, computer vision algorithms have been tasked with aiding operators by flagging potential threats (Miao et al., 2019; Akcay et al., 2018).

This scenario led to an influx of novel X-ray datasets featuring concealed threats for testing such algorithms (Mery et al., 2015; Miao et al., 2019; Isaac-Medina et al., 2021). SIXray (Miao et al., 2019) is perhaps the most renowned such dataset. It contains multiple distinctive threats that can appear simultaneously and exhibit unique properties representing the domain's real-world data.

Some of these properties stem from the penetrative nature of X-rays, such that some objects might be obscured by other benign ones, or multiple objects of interest can be stacked on top of each other (see example in Figure 1b) (Wang et al., 2021).

For instance, Hierarchical refinement (Miao et al., 2019) and Selective Dense Attention Network (Wang et al., 2021) employ innovative attention mechanisms to negate these occluding effects. Alternatively, additional pre-processing aims to aid models in "seeing" concealed objects by separating the different layers of the image (Mery et al., 2015). Although here we do not apply such techniques, they are compatible with MEOW; hence they can be utilised to try to improve our results further.

SIXray is a publicly available security dataset comprising more than a million X-ray scans of baggage with four important properties: 1. Overlapping objects 2. Inherent intra-class variation 3. Noise from the heavily cluttered objects, and 4. Heavy class imbalance (Miao et al., 2019).

Recent studies in threat detection employ standard CNNs (Hassan et al., 2020), while others utilise conditional GANs, such as Akcay et al. (2018) DCGAN-based encoder-decoder-encoder pipeline for X-ray anomaly detection. However, these approaches may be suboptimal for non-visual media like X-rays due to inherent medium properties and heavy imbalance in favour of benign samples (Mery, 2015; Dumagpi and Jeong, 2020; Miao et al., 2019). Overall, developing new X-ray-specific models is time-consuming and isolating research from visual domain advancements may hinder the joint progress of the fields. AutoML offers a potential solution (Xue et al., 2019). In particular, NAS approaches are neither bound to the specific data nor to architectural paradigms used in visual datasets. Unfortunately, many state-of-the-art algorithms require significant computational resources and are only viable for small datasets like MNIST and CIFAR-10 (He et al., 2021; Real et al., 2017; Lu et al., 2018; Wang et al., 2021). To address this caveat, RAMOSS (Resource-Aware Multi-objective approach for Semantic Segmentation, Dimanov et al. (2022)) enabled a streamlined integration with massive datasets using a hybrid metaheuristic algorithm based on evolutionary algorithms similar to Real et al. (2017). A particularly useful characteristic of Dimanov et al. (2022) (that we leverage) is its flexible encoding of architectures which uses a flat vector to determine all architecture parameters of the network and reconstruct a binary upper triangular matrix to define all the layers' connections (Dimanov et al., 2022). RAMOSS is based on NSGA-II, a multi-objective optimisation algorithm used in neural architecture search to generate a set of solutions that approximate the Pareto front (Lu et al., 2019). It employs non-dominated sorting to rank the solutions, crowding distance calculation to maintain diversity, and genetic operators to create new solutions for the next generation. Its final output is a set of non-dominated solutions that represent a trade-off between the conflicting objectives (Dimanov et al., 2022)

However, like most NAS algorithms (Real et al., 2017; Tan and Le, 2019; Zhou et al., 2020; He et al., 2021), RAMOSS spends most computational resources on training and evaluating architectures, making it suboptimal. Recent works, such as NASWOT (Neural Architecture Search Without Training, Mellor et al. (2021)) and SYNFLOW (Iterative Synaptic Flow Pruning, Tanaka et al. (2020)), offer performance approximation proxy scores to reduce computational costs, providing accurate estimates despite imperfect correlation with trained performance (Abdelfattah et al., 2021). In the present study, we combine neuroevolution NAS based on RAMOSS with proxy scores to devise a new approach which showcases the feasibility of NAS in concealed weapon detection and, more broadly, its scalability to real-world problems.

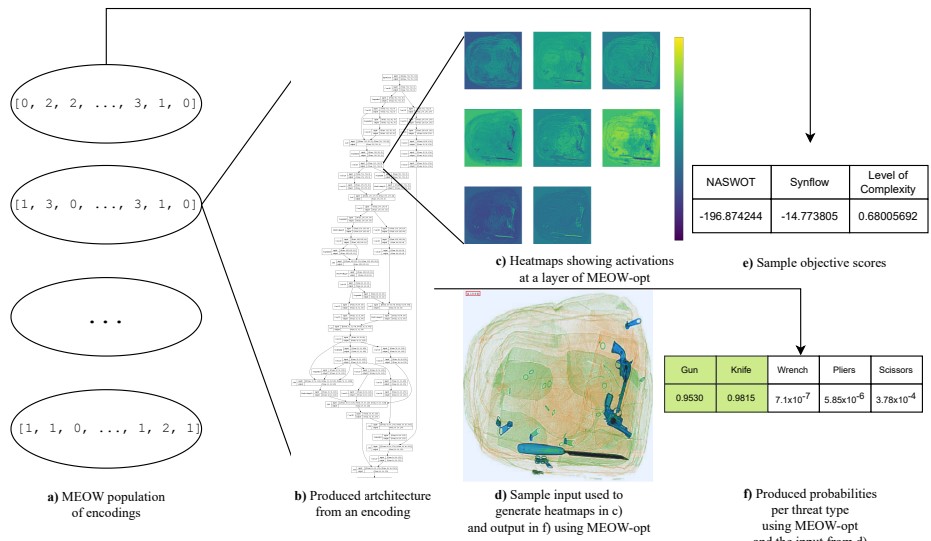

**a)** MEOW population of encodings

**b)** Produced artchitecture from an encoding

**c)** Heatmaps showing activations at a layer of MEOW-opt

**d)** Sample input used to generate heatmaps in c) and output in f) using MEOW-opt

**e)** Sample objective scores

**f)** Produced probabilities per threat type using MEOW-opt and the input from d)

Figure 2: The MEOW algorithm. The population of architectures are encoded using integer vectors (a) and then they are decoded to architectures using Dimanov et al. (2022) decoding (b). The panel shows a condensed version of the MEOW-opt architecture for SIXray. (c) Activations of a selected layer of the trained MEOW-opt when a sample input (d) is passed through the network. (e) Presents a sample objective scores of an architecture in the population, which are used for selection. Finally, (f) shows the raw predictions of MEOW-opt for input (d).

## 3 MEOW Methodology

Inspired by Dimanov et al. (2022) encoding and algorithm, here we design a similar neuroevolution process based on NSGA-II, which we upgrade with newly surfaced proxy scores (Abdelfattah et al., 2021; Mellor et al., 2021; Tanaka et al., 2020) to foster its feasibility for concealed weapon detection and its scalability to real-world settings. The encoding vector for the architectures includes parameters for each layer that determine if it is enabled, the kernel size, the operation (convolution or pooling), activation functions, and an integer value that represents a row of the binary upper triangular matrix indicating how it is connected to other layers (Dimanov et al., 2022).

In addition to the multi-label SIXray, we benchmarked MEOW on a proprietary dataset containing different threats and modifications, generated using an award-winning multi-staged algorithm to match and identify anomalous electronic devices. In this dataset, we focus on a residual X-ray image which is produced as an intermediary step of a model as input as opposed to the raw pseudo-coloured X-ray scans in SIXray, and hence we termed it "Residuals". In this proprietary dataset, "modification types" refer to different ways that objects of interest in an image are altered e.g. when items are added, removed, or tampered with. However, we are unable to provide specific examples of the modification types we are working with, due to confidentiality agreements with our industry partner. Next, using this dataset, we tested how the current state-of-the-art performs in multi-output scenarios where the model is requested to simultaneously predict the type of modification and identify what type of threat (if any) is present.

However, the computational cost of current NAS approaches impedes their direct application to threat detection in these large datasets. Thus, motivated by the promising results of "Zero-cost NAS" Abdelfattah et al. (2021), we devised a new multi-objective neuroevolution approach capable of finding optimal architectures in *less than an hour* on an RTX 3090 GPU. Specifically, we used two proxy scores - NASWOT (Mellor et al., 2021) (labelled *jacob_conv* in Abdelfattah et al. (2021)) and SYNFLOW (Tanaka et al., 2020). In contrast to Zero-cost NAS, MEOW optimises for both proxy

scores simultaneously. In short, to convert the NASWOT and SYNFLOW scores from a pruning metric to a performance-estimation proxy for assessing a specific architecture, we reformulate them like in Abdelfattah et al. (2021).

The overall NASWOT score ($NS$) per architecture is thus estimated as $NS = \ln\left(|\det(K)|\right)$ where the symmetric matrix $K$ of dimensions $n \times n$ ($n$ = number of images for the current input mini-batch $\mathbf{X}$) represents the similarity between inputs for a given $l$-layered architecture to be assessed (see details in Mellor et al. (2021)). Its entries $K_{i,j}$ are binary values characterising each pair of input images $i, j$. The following score calculation only considers layers of rectified units by the $l^{th}$-layer activation function $f_l$, and hence the output (softmax-activated) layer is excluded. The underlying rationale of the approach is that the more dissimilar the binary codes associated with each pair of inputs $i, j$ are, the easier it is for the network output layer to discern between them ((Mellor et al., 2021)). Such binary codes are computed as:

$$K = \sum_{l=0}^{L} \left( \left( g_l(\mathcal{X}) \cdot g_l(\mathcal{X})^T \right) + \left( (\mathcal{I}_l - g_l(\mathcal{X})) \cdot (\mathcal{I}_l - g_l(\mathcal{X}))^T \right) \right) \tag{1}$$

where $\mathcal{X}$ is the input to this layer ($\mathcal{X} \equiv \mathbf{X}$ for $l = 1$) and $g_l$ is a binary mapping $g_l : \mathcal{R}^+ \cup \{0\} \longrightarrow \mathcal{Z}_2$ operating over each single output of the layer $l$ such that,

$$g_l(x \in \mathcal{X}) := \begin{cases} 1 & \text{if } f_l(x) > 0 \\ 0 & \text{otherwise.} \end{cases} \tag{2}$$

We denote the full tensor containing mappings for layer $l$ outputs as $g_l(\mathcal{X})$. This tensor is flattened such that the product $\hat{K}_l := g_l(\mathcal{X}) \cdot g_l(\mathcal{X})^T$ is a $n \times n$ matrix. Likewise, $\mathcal{I}_l$ is a tensor of ones of the same dimensionality as $g_l(\mathcal{X})$. Intuitively, equation (1) computes the similarity between each pair of images encoded in binary code $\{0, 1\}$ aggregated over layers $1, \dots N$. The final $NS$ score summarises this similarity for the entire architecture by computing the log-determinant of the encoding matrix $K$.

The next proxy score considered is SYNFLOW, a *data-agnostic* (training-independent) index which focuses on weight values (instead of the activations like NASWOT), and hence it provides a complementary view to estimate the architecture performance.

In brief, the network is maximally stimulated with a responsive input (a "white" image -tensor of ones). In this setting, a low dependency of the network output (at layer $l = N - 1$) on weight magnitudes acts as a proxy for a high discrimination capability of the architecture (Abdelfattah et al., 2021). Intuitively this proxy is indicative of the importance of parameters (weights) in determining the network output, such that sparseness in relevant connections leads to more selective paths of information processing fed to the classification layer $N$ (see details in Tanaka et al. (2020)).

The score $s_l(\theta)$ per-connection (per-weight parameter) $\theta$ at layer $l < N$ is termed synaptic saliency (Tanaka et al., 2020), which is simultaneously computed for all layer connections $\theta \in \Theta_l$ as

$$S_l(\Theta) = \frac{\partial \mathcal{L}}{\partial \Theta} \odot \Theta, \tag{3}$$

where $\mathcal{L}$ is the *pseudo-loss* function, consisting of the aggregated activation for all units in layer $N - 1$ (Tanaka et al., 2020), $\odot$ is the Hadamard (element-wise) product and $S_l(\Theta)$ is a tensor of synaptic saliencies for all layer connections (note that the derivative in equation 3 is computed over each input weight $\theta$ of the layer).

Next, following Abdelfattah et al. (2021), we aggregated all individual scores per neuron $i$ in layer $l$, where the subset $\Theta_{i,l} \subset \Theta_l$ contains the neuron's input weights, to obtain $S_{i,l} = \sum_{\theta \in \Theta_{i,l}} s_l(\theta)$. To conclude, we take this one step further by averaging these summarised scores per layer and neuron, $S_{i,l}$, into a scalar index characterising the entire architecture as:

$$ln\left(\frac{\sum_{l=1}^{N-1}\left(\frac{1}{(\Theta_{l,i})}\cdot\sum_{i=1}^{(\Theta_{i,l})}ln(S_{i,l}+1)\right)}{N-1}\right)\qquad(4)$$

Preluding the main experiments, we conducted an ablation study to explore different backbone architectures and to determine which ones operate optimally for SIXray (and why). This can grant us an understanding of what design patterns should be included in the search space for optimal architectures. The study will also establish baseline models to compare with MEOW models. This ablation study is designed (1) to compare the performance of different architectures/layer blocks and (2) to provide insights into additional objectives we can consider when conducting the multi-objective optimisation.

## 4 Experimental setup

MEOW pipeline consists of five steps: (1) Specify hyperparameters, which include reference points for each of our objectives. We empirically choose the reference points for SIXray and the Residual dataset to be 600 for NASWOT and 60 for SYNFLOW score. The maximum number of convolutional layers in all experiments is 35. (2) Run the neuroevolution algorithm for $g$ generations with a population size $p$. (3) Train the produced architectures for 50 epochs following standard state-of-the-art guidelines to estimate the Pareto front of produced solutions. (4) Construct an ensemble model that combines predicted probabilities of all discovered architectures to make a final decision. (5) Comparatively evaluate all architectures, including the ensemble model. We report the best-performing architecture score from the MEOW generation (labelled "MEOW-opt" in Tables 1 and 2) as well as the score from the ensemble models (labelled "MEOW-ens" in Tables 1 and 2).

We used an Adam optimiser with a learning rate of 0.001, $\beta_1 = 0.9$, $\beta_2 = 0.99$, $\epsilon = 10^{-8}$ and a conventional categorical cross-entropy loss. Top layers are discarded, and the same five dense layers responsible for classification are added to all competing architectures to ensure a fair comparison. Along these lines, we use categorical cross-entropy as loss and also shuffle the dataset using the same random seeds. We did not use augmentation during our study to keep as many control variables as possible.

In our experiments we optimise for three different objectives which are (1) NASWOT score, (2) SYNFLOW score and (3) Level of complexity (as described in (Dimanov et al., 2021)) which measures the model's complexity in terms of number of parameters.

## 5 Results and Discussion

Results show the average precision for SIXray to facilitate the comparison with the original paper (Miao et al., 2019); and the accuracies of the two outputs for the Residuals dataset.

**SIXray benchmarks**. We start with a preliminary study of the performance of the current state-of-the-art architectures on the SIXray. Table 1 shows that, from existing staple models in the literature, the best ones (excluding MEOW) are ResNet50 and ResNet101, which are mostly on par with DenseNet. Most architectures display a reasonably consistent performance throughout "Gun", "Knife", and "Pliers" (the threat class with the most samples) classes; however, performance drops substantially for "Scissors" (the most undersampled one) and "Wrench" classes (Figure 1b). Further analysis shows a sustained increase in performance from ResNet34 to ResNet50, followed by a consistent drop from ResNet50 to ResNet101. This phenomenon can be explained by the depth of ResNet101, which may prevent it from converging with our training configuration. It is also possible that the model is overparameterised for this particular problem in line with the findings of Pasupa and Sunhem (2016) and Malach and Shalev-Shwartz (2019).

Interestingly, models produced by the new MEOW approach effectively captured the imbalanced nature of the data, as demonstrated by their consistently higher performance in all classes -including

Table 1: SIXray-10 results. MEOW-opt and MEOW-ens methods achieve the highest mAP by a significant margin. For "Pliers", the second highest oversampled class, DenseNet achieves the best performance, but MEOW architectures balance the rest of the classes more effectively and are especially successful at classifying the most underrepresented class "Scissors".

| Architecture | AP Gun | AP Knife | AP Wrench | AP Pliers | AP Scissors | mAP |
|---|---|---|---|---|---|---|
| ResNet34 | 89.71 | 85.46 | 62.48 | 83.50 | 52.99 | 74.83 |
| ResNet50 | 90.64 | 87.17 | 64.31 | 85.78 | 61.58 | 77.87 |
| ResNet101 | 87.65 | 84.26 | 69.33 | 85.29 | 60.39 | 77.38 |
| Inception-v3 | 90.05 | 83.80 | 68.11 | 84.45 | 58.66 | 77.01 |
| DenseNet | 87.36 | 87.71 | 64.15 | **87.63** | 59.95 | 77.36 |
| MEOW-opt | 94.93 | 89.47 | 67.48 | 86.13 | 89.29 | 85.46 |
| MEOW-ens | **95.51** | **94.04** | **77.34** | 76.12 | **96.34** | **87.87** |

Table 2: Residuals dataset results. Columns show averaged accuracy for two separate multi-class classification outputs: (1) detecting the type of modification and (2) identifying the threat associated with this modification. MEOW discovered architectures outperform the state-of-the-art, even though they are remarkably more parsimonious than the rest.

| Architecture | Threat Accuracy | Modification Accuracy |
|---|---|---|
| Inception-v3 | 77.52 | 79.21 |
| Nasnetmobile | 68.15 | 72.96 |
| Inceptionresnetv2 | 89.06 | 88.22 |
| Resnet50 | 88.82 | 87.26 |
| MEOW-opt | 89.18 | 88.34 |
| MEOW-ens | **91.23** | **89.06** |

Table 3: Residuals results in % of F1 score per class. MEOW architectures, consistently with SIXray results, achieve the best performance (except for one type of modification, and by less than 0.5%); despite MEOW architectures being substantially smaller than all state-of-the-art models.

| Architecture | Threat | | | | | Modification | | | |
|---|---|---|---|---|---|---|---|---|---|
| | Threat 1 | Threat 2 | Threat 3 | Threat 4 | None | Type 1 | Type 2 | Type 3 | Benign |
| InceptionV3 | 87.46 | 87.80 | 80.36 | 70.00 | 71.00 | 91.13 | 83.94 | 60.08 | 71.30 |
| NasnetMobile | 72.22 | 84.52 | 40.26 | 67.06 | 67.65 | 86.01 | 77.20 | 51.72 | 66.95 |
| InceptionResNetv2 | 89.03 | 89.76 | 85.71 | 90.28 | 89.60 | 93.54 | 87.88 | 78.19 | **90.06** |
| ResNet50 | 89.70 | 89.55 | 88.70 | 89.07 | 87.33 | 92.49 | 88.00 | 77.02 | 86.96 |
| MEOW-opt | 88.74 | 87.82 | 87.55 | **90.40** | 90.06 | 93.78 | 87.66 | **79.89** | 88.24 |
| MEOW-ens | **92.11** | **92.94** | **91.98** | 90.27 | 90.00 | **94.81** | **88.02** | 79.53 | 89.74 |

the undersampled ones (Table 1). This suggests that the MEOW's multi-objective optimisation employed for architecture search renders highly-discriminative models. This property can be attributed to the proxy scores since they are designed to reward architectures that can discern dissimilar features in the data. In this case, it resulted in models that were particularly effective at differentiating between the various classes. In short, the best MEOW architecture outperforms state-of-the-art models on the SIXray dataset overall, even when compared to class-balanced hierarchical refinement (CHR, (Miao et al., 2019)) enhanced architectures.

The higher performance of MEOW architectures w.r.t. the state-of-the-art calls for further investigation into the importance of designing domain-specific architectures. Remarkably, the optimal MEOW-opt architecture utilises just above 11 million parameters, significantly fewer than

Table 4: Results on Residuals dataset for semantic segmentation tasks. Columns highlight the ( first , second and third best) approach. While MEOW-opt does not achieve the best test *IOU* and *F*1, it remains the only architecture capable of yielding a validation performance which can resemble its performance on an unseen dataset. In addition, MEOW uses only 0.53*M* parameters, compared to 62*M* and 30*M* for the other slightly higher-performing alternatives making it far more favourable w.r.t. computational efficiency.

| Architecture | Number of parameters (M) | Validation IOU (%) | Validation F1 (%) | Test IOU (%) | Test F1 (%) |
|---|---|---|---|---|---|
| MobileNet - UNet | 8.34 | 53.27 | 69.51 | 21.60 | 35.52 |
| VGG16 - UNet | 23.75 | 55.49 | 71.38 | 33.48 | 50.16 |
| ResNet18 - UNet | 14.34 | 54.96 | 70.93 | 37.98 | 55.05 |
| ResNet34 - UNet | 24.46 | 64.46 | 78.39 | 37.73 | 54.79 |
| ResNet50 - UNet | 32.56 | 53.09 | 69.36 | 39.18 | 56.30 |
| InceptionResNetv2- UNet | 62.06 | 62.93 | 77.25 | 42.80 | 59.95 |
| Inceptionv3 - UNet | 29.93 | 59.49 | 74.60 | 43.98 | 61.09 |
| MEOW-opt | 0.53 | 40.04 | 57.19 | 40.56 | 57.72 |

the 23.5 million parameters used by ResNet50. Thus, we will focus next on identifying the properties of MEOW discovered architectures that underlie its high performance by establishing parallels with the CHR approach (Miao et al., 2019). One of the key CHR steps is the utilisation of low- and high-level features for refinement by concatenating intermediate activations and then filtering out noisy information based on the signals from the next layer. Subsequently, each layer $l$ selected for feature extraction becomes a separate stream of activations $a^l$ fed into an auxiliary classifier $y^l := f^l(a^l, \xi^l)$ where $\xi^l$ is the hashing vectoriser of the selected layer and, finally, the expectation across layers provide class-probability outputs $y$. Along these lines, some large skip connections are also observed in the architecture discovered by MEOW, reminiscent of the CHR effect. After the first convolutional layer, a skip connection links to a much lower dimensional representation layer. Then, signals are added in what resembles an auxiliary feature extraction arm in (Miao et al., 2019), which is effectively similar to what the CHR accomplishes with separated streams. Noticeably, the architecture seems to be composed of ResNet-like blocks with various degrees of skip connection depth, which, as we can see from Table 1, seems to be working better than InceptionNet or DenseNet-like blocks.

In an attempt to utilise a better portion of the generated front of solutions rather than just one (in the form of MOEW-s1), we decided to select four architectures with the highest contributing hypervolume. Then, we designed an ensemble classifier which combines their output probabilities $y, y'...y'''$ to compute an ensemble prediction $y_{ens}$. This ensemble model (MEOW-ens) simply incorporates an extra custom layer to compute the optimally weighted average of the four predicted probabilities and make a final prediction. In summary, MEOW-ens achieves the best overall scores, with the sole exception of the most out-of-distribution upsampled class in the testing set - "Pliers". Overall the ensemble approach outperforms the state-of-the-art by more than 10%, which is over five-fold the state-of-the-art improvement compared to the CHR approach (Miao et al., 2019).

**Propietary dataset benchmarks**. Next, we comparatively test the effectiveness of our new AutoML framework versus state-of-the-art architectures in a real-world industrial problem, the Residuals dataset outlined in Section 3.

Succinctly, each scan outputs an associated threat type and a modification type; thus, all models are fitted with the same multi-output multi-class top layers (similar to our SIXray experiments). Table 2 shows that MEOW architectures are once again outperforming all state-of-the-art architectures, in line with SIXray results. The gap between the state-of-the-art and MEOW is significantly smaller but still consistent with SIXray results. Judging by the overall results, NasNetMobile and Inception-v3 fail to capture the features of the datasets, which reveals that conventional *avant-garde* architectures should not be treated as a silver bullet to directly address an arbitrary computer vision

problem. Having a 10% drop compared to ResNet is also inconsistent with the results for SIXray. It uncovers a gap in our ability to estimate their performance on a new dataset and the benefits of tailoring an architecture to fit the particular problem.

To investigate the results more closely, we look into the $F1$ scores for each class for both of the outputs (threat type and a modification type) as presented in Table 3. Again, MEOW architectures consistently take up top performance positions. However, in contrast to SIXray, the capability of the ensemble is limited since the best MEOW architecture (MEOW-opt) outperforms the ensemble method for some of the classes. Even though MEOW-ens achieves the best overall results, we attribute this discrepancy to lower diversity in the approximation set of MEOW for the Residuals dataset. Since we used only 20 generations with a population size of 20, it is fair to assume that the algorithm did not successfully explore the enormous search space. Future work needs to address this caveat with an ablation study over these two hyperparameters.

Strikingly, thanks to the proxy scores (Section 3), MEOW's computational time is *just under one GPU hour* for a population size of 20 and for 20 generations, which is a drastic improvement over the 8 hours required for previous, and already highly optimised, AutoML approaches such as Dimanov et al. (2022), used to discover a segmentation model on Cityscapes with the same hyperparameters as in MEOW. In addition, Dimanov et al. (2022) was already, to our best knowledge, one of the fastest NAS runs to date, and Cityscapes dataset is also three times smaller than Residuals, making the 8x speed-up of MEOW particularly remarkable. Moreover, since there is no training during the discovery phase, it is possible to run MEOW without the need for a GPU since the CPU can handle inference on most machines. This characteristic contributes to making the research field more accessible for new practitioners and also feasible to use in a plethora of new domains and industry settings, as suggested by our results.

In this study, we successfully showcased the enhanced capabilities of MEOW to tackle multi-input/multi-output and multi-label classification tasks. Although MEOW is based on RAMOSS, which is originally designed to address semantic segmentation problems, it demonstrated its versatility by adapting to these new challenges.

To further explore its potential, we utilised a supplementary task from the Residuals dataset, redefining the problem as a semantic segmentation task while maintaining the same multi-input sequence (i.e., two input images provided to the network to calculate the residual). The next experiment is specifically designed to test if the expected loss in performance is minimal and compensated by key advantages such as resource economy and better generalisability.

Table 4 compares the performance of MEOW with much larger state-of-the-art methods in semantic segmentation for the Residuals dataset. Results from the segmentation study are consistent with the ones of RAMOSS from Dimanov et al. (2022). The best-balanced MEOW architecture (MEOW-opt) manages to achieve the third-best results in terms of test IOU and F1 score while being *30 to 60 times smaller* in terms of the number of parameters compared to the first and second methods. Notably, MEOW-opt is the only approach that shows strikingly similar results between the validation and the testing sets. That is, the generalisation error is both low and more representative of its real-world performance compared to competing alternatives. In contrast, results highlight the potentially detrimental effect of using a well-performing generic architecture like, e.g., ResNet34-Unet when it is applied in a real-world scenario. We speculate that this consistent performance is tied to the fact that MEOW-opt is not overparameterised like some of the alternatives, although the second least parameterised architecture (MobileNet-UNet) seemingly fails to display this ability.

To conclude, although the causality of the consistent performance of MEOW has yet to be established, we attribute its generalisation capacity to the fact that the neuroevolution discovery process adapts to the problem at hand, generating naturally tailored architectures. This serves as further evidence for the effectiveness of the neuroevolution-controlled optimisation and suggests the need for further research on AutoML techniques for industrial problems.

## 6 Conclusion and Future work

Traditional neural architecture search, a particularly fast-evolving, "hot" research topic, has limited use in the industry, given its extremely high computational cost. In this study, we propose a new NAS approach to feasibly address real-world problems in threat detection, strongly reducing computational demands. In addition, from a practical perspective, the AutoML approach is modularised and hence streamlined to be used with new datasets, unlike other AutoML techniques (e.g., Real et al. (2017); Liu et al. (2018, 2019)).

In this study, we designed an ensemble approach that leverages multiple sub-optimally discovered architectures instead of disregarding them (as it is typically the case). Although the ensemble strategy provides promising results, future work can explore using the collective knowledge of the generated networks to conduct *knowledge distillation* (Hinton et al., 2015). The presented method typically improves the overall performance of the state-of-the-art in both datasets used. Specifically, its behaviour in the most extensive public concealed weapon detection dataset (SIXray) to this date avows the importance of making AutoML scalable to real-world scenarios; and designing such systems in a "disentangled" fashion from the dataset used as a proof-of-concept.

Results in both datasets suggest that the heuristic performance estimation such as the one implemented in MEOW [1] can drastically reduce the computational cost of such algorithms, effectively replacing highly resource-greedy evaluation in these settings. Here, we also modify a well-performing multi-objective approach to use multiple proxy scores to speed up the architecture search and showcase how these proxies can be used in conjunction with multi-objective optimisation to outperform state-of-the-art architectures. In summary, the results and the remarkably low search time (about an hour) without the need to use GPUs provide further evidence of the untapped potential of AutoML in industrial applications.

To conclude, albeit obvious, it is worth stressing that understanding how a model operates is entangled with the understanding of the dataset used to train and evaluate it (giving birth to the popular saying in the deep learning community "garbage in, garbage out" (Kilkenny and Robinson, 2018; Rose and Fischer, 2011; Sanders and Saxe, 2017)). Along these lines, future work should explore the idea of repurposing our AutoML efforts to make a contribution to data quality.

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

## A CIFAR-10 results

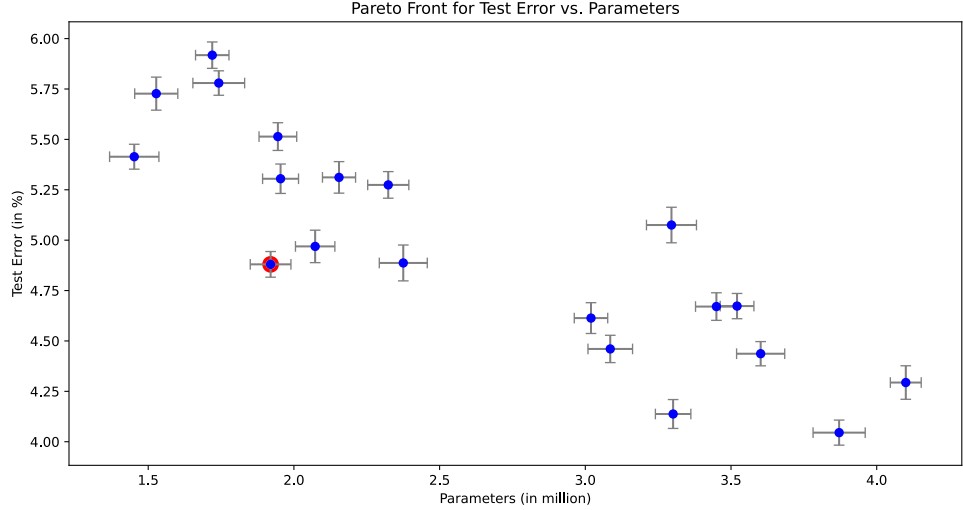

Figure 3: The Pareto front of generated solutions with error bars for the test error and the number of parameters for CIFAR-10. The red-marked sample represents the optimal encoding which has the highest contributing hypervolume, referred to throughout this study as "MEOW-opt".

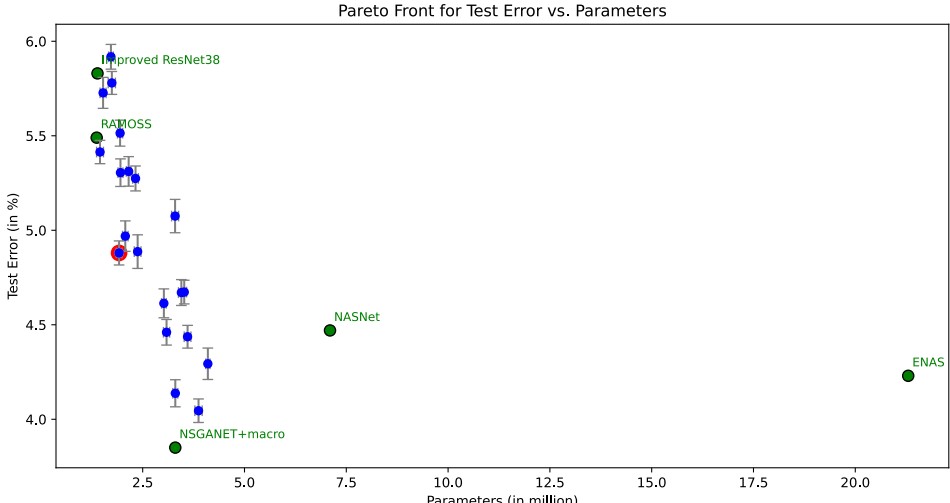

Figure 4: The Pareto front of generated solutions for CIFAR-10 with error bars for the test error as a function of the number of parameters. Some state-of-the-art methods are shown for comparison (green dots).

Here we present the results of a 10-run case study on the CIFAR-10 dataset, which is a widely-used benchmark for neural architecture search. The section includes a Pareto front of generated solutions (Figure 3). Additionally, we compare our approach to state-of-the-art models, shown in a separate figure (Figure 4) for clarity. Results further support the generalisability of our approach beyond threat identification problems, which is the main focus of this study. As expected,

specifically-designed architectures for CIFAR-10 can provide lower test errors than the novel approach presented. However, MEOW-opt (red marked solution in Figures3 and 4 consistently provides the best trade-off when compared to other methods in terms of balancing the different objectives, that is, it is associated with the highest contributing hypervolume.

## B  Search space used

Table 5 presents the search space used to conduct the experiments in this study. It is important to note that all the specified hyperparameters to filter the search space are adjustable.

| Parameters | Range |
|---|---|
| Active | $\{0,1\}$ |
| Num filters | $\{2^i\}_{i=1}^{log_2 256}$ |
| Num nodes | $\{2^i\}_{i=1}^{log_2 256}$ |
| Kernel | $\{3,5,7\}$ |
| Batch Normalisation | $\{0,1\}$ |
| Dropout | $\{0,1\}$ |
| Pooling | $\{0,1,2\}$ |
| Connections | $\{i\}_{i=1}^{2^{(n_c+n_d)}}$ |

Table 5: Layer parameters defining the search space. $n_c$, $n_d$ are the number of convolutional and dense layers, respectively. For each convolutional layer, all of the following are specified. Values and significance of these parameters are discussed in more detail in (Dimanov et al., 2022)

## C  MEOW SIXray architecture

Figure 5 displays the MEOW-opt architecture used for SIXray (split into two pages). This visualisation also truncates some of the top and bottom layers of the architecture to enhance legibility. The exported version of the architecture is available in our repository `https://anon-github.automl.cc/r/MEOW-B852`.

Figure 5: MEOW-opt architecture for SIXray (truncated after the last convolution to enhance legibility).

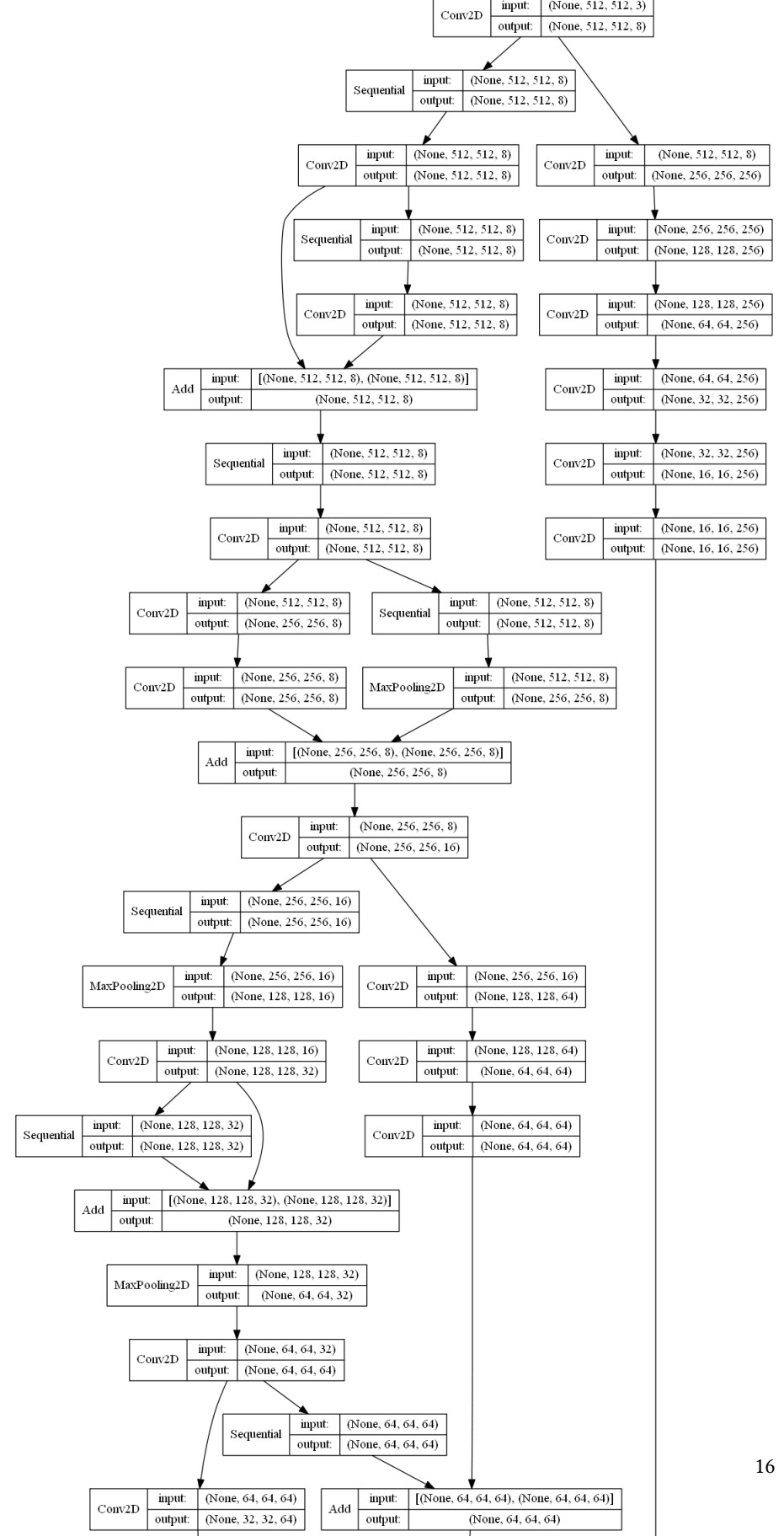

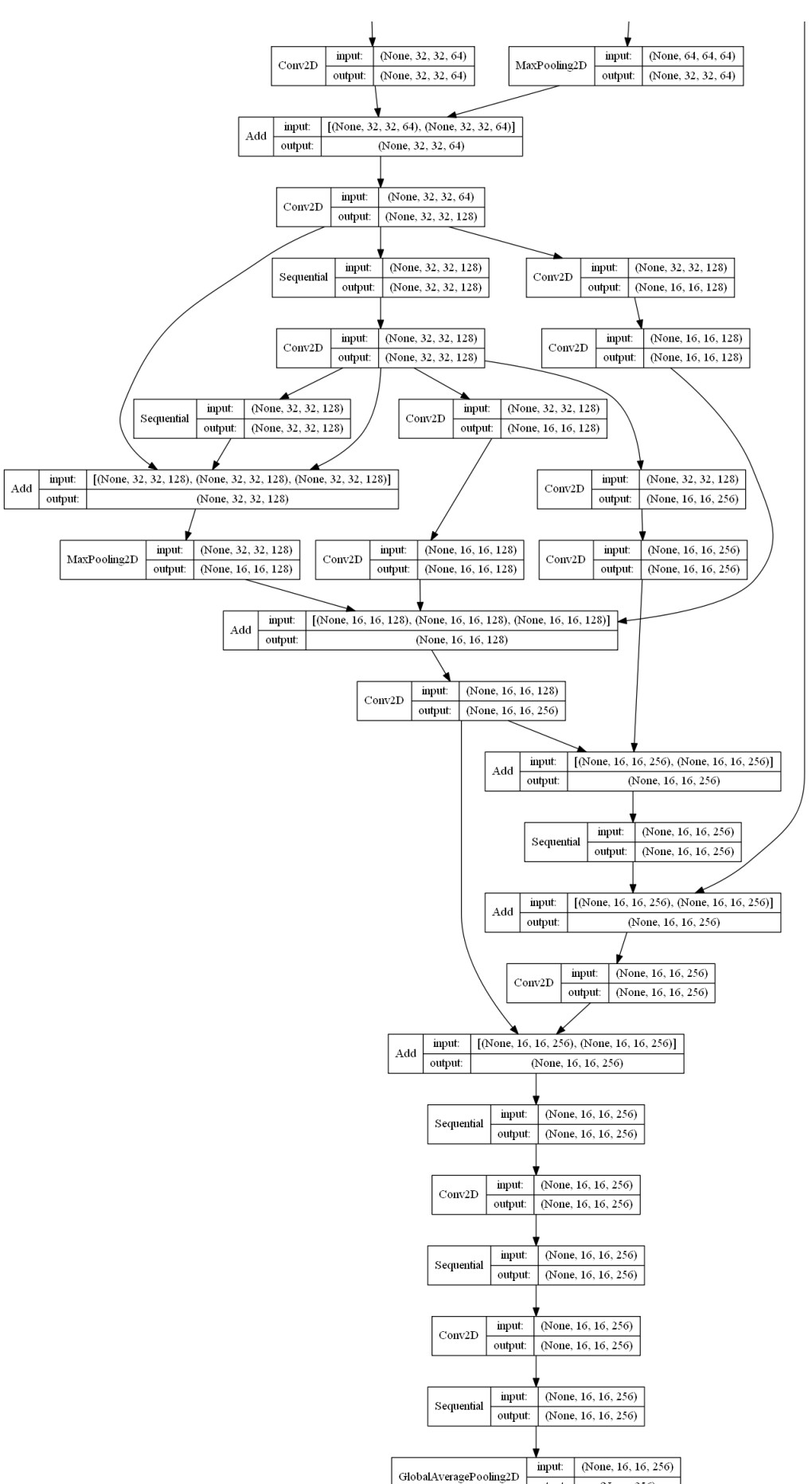

## D  Broader Impact Statement

This research presents MEOW, a novel automated machine learning (AutoML) multi-objective approach for neural architecture search (NAS) in concealed weapon detection problems. The proposed method offers significant benefits, such as reduced computational costs, increased adaptability, and improved security measures in crowded public areas. However, potential risks, such as misuse by malicious actors and overreliance on automated systems, must also be considered. The development and adoption of MEOW could lead to a safer society and increase public trust in security measures, but it is essential to implement proper regulation and oversight to ensure responsible and ethical use.

Uncertainties surrounding the technology include the long-term impact on privacy and civil liberties, as well as its scalability and adaptability to new and evolving threats. The results of this study showcase the untapped potential of AutoML in industrial applications and open up new avenues for research and development in various sectors. However, focusing on both the positive applications and potential negative consequences is crucial when discussing the broader impact of this research.

In conclusion, MEOW has the potential to significantly improve security measures in public spaces and save lives by detecting concealed weapons. Nevertheless, it is essential to balance the positive outcomes with the potential risks and uncertainties associated with the technology. This requires a thoughtful approach that considers the ethical implications, societal impact, and potential misuse of the technology while working towards creating a safer and more secure environment.

