# OpenReview forum: "MEOW - Multi-Objective Evolutionary Weapon Detection"
_automl.cc/AutoML/2023/Conference — AutoML 2023 MainTrack_

### Official Review · Reviewer_SdUM · 2023-04-11

**Potential Impact On The Field Of Automl Rating:** 3
**Technical Quality And Correctness Rating:** 3
**Clarity Rating:** 2

**Summary Of Contributions:**

The paper proposes a new method to generate neural network architectures for the task of concealed weapon detection from X-ray images.
It combines multiple efforts from computer vision, first by adopting a neural architecture search procedure that doesn't involve training the model being evaluated and thus saving a considerable amount of time.
Second, the paper uses different methods of assessing the quality of the evaluated model, making up a multi-objective optimization problem.
This optimization problem is solved with an evolutionary search strategy where each input encodes the architecture of the evaluated model.

Further, in order to make use of different solutions found during the optimization phase, an ensemble is built from the top performing models, which avoids over-committing to a specific model.
The experiment section shows the proposed method performs well on two weapon detection tasks, especially better than other benchmarks on rare instances depicting uncommon weapons.

**Actions Required To Increase Overall Recommendation:**

More clarity about the evaluation metrics being used in the experiment section will help the paper.
I will appreciate more intuition building in Section 3, maybe via a small illustration, where the different objectives are explained.

Ablation studies for the multi-objective aspect will make me get a better insight.
Does optimizing individual objectives work just as well?
Can we do better than picking out the top solutions during optimization to build an ensemble?

**Clarity:**

The clarity of the paper could be further improved.
When reading the paper, I often find myself not understanding the details that are at the center of the proposed algorithm.
For example, some discussion could be devoted to NSGA-II and why it's appropriate for this kind of optimization task.

Section 3 explains the math behind the various objectives used in the multi-objective optimization and can benefit from some illustrations and toy examples to show that these scores are effective at quantifying the quantities of interest.
For example, Eq. (4) isn't very illuminating in terms of what the score represents.

As mentioned, it's not clear what the numbers in Table 1 represent.
Table 2 shows the various accuracy metrics, but their differences, or even the difference between "threat" and "modification", the two main sections in Tables 2 and 3, were never clearly explained.

Miscellaneous typos:
- line 6: asses --> assess
- line 106: The encoding algorithm of Dimanov et al. (2022)
- line 302: their --> its

**Overall Review:**

The paper presents a promising approach to NAS for concealed weapon detection, an important problem.
The experiments show results in favor of the proposed approach and contain interesting insights.

Clarity is a concern for the paper, as a lot of details are currently missing, making it difficult to fully appreciate the proposed algorithm.
More setup is also needed to fully crystalize the different metrics being used to evaluated the models in the experiment section.
As of now, "performance" is quite a vague term that's being used liberally in the paper.

In terms of technical insights, the paper can benefit from further ablation studies that pinpoint what exactly makes the proposed method work.
Is it the multi-objective approach or simply just one of the scores being optimized?
What are the different ways of building the final ensemble and how does that affect performance?

**Potential Impact On The Field Of Automl:**

This paper tackles an important problem and proposes an interesting solution.
AutoML researchers would find value in this work.

**Review Confidence:**

3: You are fairly confident in your assessment. It is possible that you did not understand some parts of the submission or that you are unfamiliar with some pieces of related work.

**Review Rating:**

4: Weak Reject: For instance, a paper with minor technical flaws, limited impact, and/or weak evaluation.

**Review Summary:**

I think the paper contributes a reasonable approach to an important problem and presents promising results.
However, more details are needed to make the submission a strong one, and I recommend a weak rejection.

**Technical Quality And Correctness:**

The paper proposes a reasonable approach for the given problem.
The experiment section is convincing in showing that the proposed method works well compared to the included baselines.

A couple of aspects I think could be improved.
The paper doesn't seem to mention what performance metric Table 1 uses (accuracy, recall, etc.).
I would have liked to see the average results accompanied by error bars.

The paper could also spend more effort highlighting the benefits of the multi-objective optimization approach, thus clarifying its contributions.
For example, what happens when we simply optimize for each of the objectives used?
Does the discovered architecture work as well?
If not, which objective plays the most important role, or is it the combination of multiple objectives?

Finally, I agree that the ensemble approach is promising but was a bit disappointed that it's simply the top 4 performing solutions that are picked to form the ensemble.
Perhaps some diversity-aware approach could work here, for example where we select solutions that span the Pareto front as much as possible.
After all, the idea of having an ensemble is to avoid over-committing to one specific model/hypothesis.

---

> ### Author Response · Authors · 2023-05-01
> **Thank you very much for raising these points. Please see more details on the manuscript updates below, as requested.**
>
> 1. Performance metrics in Table 1: We provided a too succinct explanation of the metrics used in Table 1. The caption has been expanded in the revised version. In short, we used Average Precision (labelled "AP" in the manuscript) and mean average precision (mAP) in order for our results to be comparable to the original study of SIXray. Typically, precision is the metric used for SIXray given that most of the images are negative, and hence reducing the number of false positives is fundamental in this setting.
>
> 2. Further clarity on the advantages of the multi-objective approach: We apologise for not communicating this key point more clearly. A multi-objective approach is needed to balance the network's performance and size since the efficiency and compatibility of the produced architecture can be as important as its performance in realistic settings.
>
> We agree that ablation studies are interesting to provide an even more comprehensive view of the robustness of the approach and bring further insights. Along these lines, please note that the proposed method outperforms single-objective ones, showcasing the distinct advantage of this feasibility study.
>
> In the revised manuscript, we have highlighted these advantages of the multi-objective results, as requested (see reworked Section 5, discussion about the segmentation, stating that results show the compelling need for considering multiple objectives). In these results, it can be observed that the process suggested that it was optimal to slightly sacrifice accuracy while reducing the model size by several orders of magnitude (the optimal MEOW model is more than x50 and more than x100 times smaller than the 1st and 2nd ones, respectively), to make it much more computationally efficient and compatible with a plethora devices and of realistic industrial settings.
>
> 3. Further clarity on the ensemble approach: We fully agree that, in principle, alternative ways of combining models can also be useful. In this work, we aimed to conduct a feasibility study and establish that this might potentially be a promising research direction. As indicated, more diversity can be added for other scenarios not tested here, and we now acknowledge this in the revised version, as requested. We did not mention this aspect in the previous study since this was not needed for our benchmarks (note please that we have added a further public dataset); thank you for the input.
>
> 4. Clarity in NSGA-II and other clarifications in the Introduction: We have added extra argumentation about the need for multi-objective optimisation in the Introduction and also some very brief discussion of the benefits of NSGA-II in the Related work. Please consult the revised manuscript. Thank you for raising this point and for your advice.
>
> Thanks a lot also for bringing these two typographical errors to our attention; we apologise for any confusion they may have caused. Regarding the reference to "Dimanov et al. (2022) encoding and algorithm," we would like to please clarify that our work was inspired by the neuroevolution algorithm presented in their work, as well as the novel encoding technique used in conjunction with this algorithm. We would like to thank you very much for your useful input, and we are looking forward to your reply and discussing any further queries.

---

> > ### Comment · Reviewer_SdUM · 2023-05-03
> > **remaining questions**
> >
> > Hi authors, thanks for their responses.
> > - Regarding the precision metric, wouldn’t you care about false negatives, which are more catastrophic than false positives?
> > - Thanks for the clarification of the multiple objectives; my understanding of the paper is now better. Where exactly is the comparison between your approach and the single-objective methods? Perhaps an enumeration of what the specific objectives are and how they’re optimized would help clarify things

---

> > > ### Author Response · Authors · 2023-05-03
> > > **Thank you very much for your interest, reply, and follow-up questions.Of course, please find below the clarification to your queries.**
> > >
> > > 1. False negatives: We did not report the false negatives rate or the recall (this was not reported in the original SIXray paper either) because all recall values were above 90%, and thus, false negative rates were not very informative of the performance, due to the 10:1 ratio of negatives to positives. Moreover, in this domain, industry partners were much more interested in the false positive rate than the false negatives since current checks in many places are done at random intervals, and any system which would not disturb the throughput of items/people would be considered of great value. Our view fully aligns with yours though, all values should have been reported, we cannot edit the manuscript now (it is not allowed) but we can mention this in the camera-ready version.
> > >
> > > 2. Different objectives: We apologise for not stressing the different objectives enough. They are the following:
> > >     Objective 1: NASWOT score
> > >     Objective 2: SYNFLOW score
> > >     Objective 3: Level of complexity (number of parameters)
> > >
> > > These are shown in Figure 2. In addition, in the revised manuscript, we also added an additional paragraph at the end of Section 4. Experimental Setup. Thank you for noticing this inconsistency, and we hope this modification in the updated manuscript has resolved this issue.
> > >
> > > 3. Single objective benchmark: The reviewer makes a great point about showing the benchmark to single-objective methods explicitly. We adopted this for the revised version, shown in Figure 4 in the Appendix with our CIFAR-10 experiments, as some of the chosen reference methods are indeed single-objective. In this study, we didn't focus on single-objective methods since the main challenge in the area was on finding the optimal balance between cost and performance. However, it is perfectly feasible to adjust the underlying RAMOSS algorithm MEOW is based on such that single-objective optimisation is supported. We will complete this further benchmark and add it to the repository as requested; thank you very much for the idea. We cannot edit the manuscript further (it is locked) but rest assured that for the camera-ready version the following sentence will be added:
> > > "In our experiments, we optimise for three different objectives which are (1) NASWOT score, (2) SYNFLOW score and (3) Level of complexity (as described in Dimanov et al. 2021) which measures the model's complexity in terms of the number of parameters."

---

### Official Review · Reviewer_no5F · 2023-04-13

**Potential Impact On The Field Of Automl Rating:** 3
**Technical Quality And Correctness Rating:** 3
**Clarity Rating:** 4

**Summary Of Contributions:**

This paper introduces MEOW, a multi-objective NAS approach for weapons detection in X-ray images that are commonly used in security applications. The proposed approach combines neuroevolution NAS with proxy scores to find better-performing architectures in terms of training time, size/parameters for concealed weapons detection in X-ray images. NSGA-II is used to evolve architectures, and two complementary proxy scores (NASWOT and SYNFLOW) are modified to evaluate the performance of each architecture. These proxy scores are used as objectives within NSGA-II to generate an optimal and ensemble architecture from the pareto front. The MEOW approach results in a considerable speedup over other AutoML approaches, and results in a new ensemble and optimal architecture that are improve multi-class, multi-label mean prediction accuracy by 10% compared to popular CV architectures. The benefits of using MEOW are demonstrated on the publicly available SIXray dataset and the proprietary Residuals dataset.

**Actions Required To Increase Overall Recommendation:**

Please see comments above.


**Clarity:**

Overall, this work is presented in a very clear way that makes it easy to understand. The writing and flow is clear and logical. Most terms are well-defined, and the distinction from existing work is clear. Some suggestions for improving the clarity are:
1. A couple of sentences describing NSGA-II could further improve the clarity.
2. Produced architecture in Figure 1 is unreadable. I would also be interested too see what architectures are used in the ensemble.


**Overall Review:**

The paper is very well written and easy to follow. Concepts are explained clearly and using NAS for weapons detection in X-ray images is technically interesting. THe proposed approach combined multi-obective search with proxy scores to evaluate discovered architecture performance. Experiments on two datasets resulted in accuracy improevemts and NAS search time speedups. The authors also mention that the datasets ued for experimentations are larger than those used in related works, making the speedup remarkable. These finding have implications for the broader AutoML community in developing neuroevolution NAS approaches for specific applications.


The negative aspects of this paper pertain to the limited number of runs (the authors used 5 runs of NSGA-II). More runs could lead to deeper understanding of performance, and differences in architectures found.


**Potential Impact On The Field Of Automl:**

The approach presented in this paper is impactful because of improvements in NAS speedup on large weapons detection datasets, and improvements in accuracy. MEOW finds smaller, more effective architectures (it uses 0.5M parameters), improves accuracy by upto 10% and requires 1 GPU hour for NSGA-II to run. This approach makes it cheaper to find smaller, more effective architectures for different domains.


**Review Confidence:**

3: You are fairly confident in your assessment. It is possible that you did not understand some parts of the submission or that you are unfamiliar with some pieces of related work.

**Review Rating:**

8: Accept: Technically sound paper with major impact and strong evaluation, with perhaps some minor flaws.

**Review Summary:**

This paper presented an approach that used multi-objective search with proxy scores as objectives for NAS. The paper also demonstrated the effectiveness of MEOW on a relatively new problem - that of concealed weapons detection in X-ray images. The results improved accuracy by 10% and led to an atleast 8x speedup compared to existing CV architectures. Furthermore, the datasets used for evaluations were larger than those used in related work.


**Technical Quality And Correctness:**

The experiments, theory and experiments are generally of sound quality. Perceived flaws include:
1. 5 runs may not be enough for a statistically significant result. Considering that it takes 1 GOU hour for one NSGA-II run, it could be better to incorporate some more runs.
2.  It could be beneficial to include the error bars on each NSGA-II run as well, for a clearer idea of algorithm performance.


Minor nits: I’m interested to know if the objectives are conflicting or complementary as mentioned in Section 3. It is also not clear to me how the 200x speedup mentioned in Section 1 was calculated. I’d also be interested in analyzing the pareto fronts generated by NSGA-II in terms of the objectives.

---

> ### Author Response · Authors · 2023-05-01
> **Thank you for the detailed input and for your interest in our study. Please find below our answers to your comments.**
>
> 1.  More number of runs: We fully agree with the need for computing as many runs as possible to draw reliable, statistically significant conclusions. To this end, we have now benchmarked our approach with an extra public dataset for 10 runs instead of 5, as requested. This was also done to further foster reproducibility on another public dataset, even beyond weapon detection. Given the limited time to address the points in this rebuttal, this was the only feasible public dataset to complete the experiments in time across more than five runs and in line with other reviewers' requests. Please find them in the updated Appendix A (Section 7 currently).
>
> The reason why we did not present further runs in the previous submission was two-fold:
> First, the realisation that all runs provide very similar results for all datasets (less than 1\% variance in the scores). Thus error vars (either defined by variance or standard errors of the mean) were hardly visible. Indeed with more time ahead, we can provide well over 10 runs for all datasets. Thus this is precisely a distinctive advantage of our approach w.r.t previous models: the feasibility of comprehensively employing AutoML for industrial datasets.
>
> Also, the NAS process takes 1 GPU hour, but the retraining of the architectures also takes around the same or even more, depending on the dataset. Hence, in real-world settings (the scenario this approach is designed for), it takes 2-3 GPU hours to find the architecture and benchmark it from scratch (for selecting the one with the highest contributing hypervolume).
>
> However, please note that we need to retrain all produced architectures from all runs to faithfully compare them, which will require 200x (given a population size of 20 and 10 runs) more time than in a real setting. That is why, and simply due to the limited time for drafting the rebuttal and updating the study, we have focused on rerunning and including 10 runs on CIFAR-10 for this rebuttal (instead of some larger dataset).
>
> We would also like to note that the scope of this feasibility study was to present a proof-of-concept to challenge the (sometimes implicit) hypothesis that AutoML neuroevolution NAS methods cannot outperform expert-designed ones by showing otherwise in some key use cases.
> We thank you very much for highlighting this key aspect; we will stress this distinctive advantage more, as indicated in this and in all follow-up studies.
>
> 3. Clarification of the 200x speedup. Thank you for spotting the missing explanation behind the number, and we apologise for it. We have added a reference to the works we talk about, and these two works are NSGANET (Lu et al., 2019) which uses 8 GPU days (192 GPU hours) and SqueezeNASNet (Shaw et al., 2019), which uses between 11 and 15 GPU days for discovering an optimal architecture.
>
> 4. Couple of sentences describing NSGA-II. Sorry for the omission (for space limitations). We have now added some additional sentences to the beginning of the methodology to explain NSGA-II briefly.
>
> 6. Legibility of the optimal architecture. We agree that the architecture is not legible. Thank you, we have now included a much larger version of it in Section 9, which will become Appendix C (Section 9 currently). Visualisations and exports of all used architectures (including the ones for the ensemble) will be uploaded to the repository.
>
> Thank you again for your interest in our study and for your input; we are looking forward to hearing your views and any further requests you may have.

---

### Review · Reproducibility_Reviewer_cDeV · 2023-04-20

**Completeness Of Code And Dataset Supplement Rating:** 4
**Usability And Ease Of Reproducibility Rating:** 4

**Actions Required To Increase The Reproducibility And Overall Recommendation:**

Improve the readme
annotate the notebooks

**Completeness Of Code And Dataset Supplement:**

I could only find the ResNet 34 and 50 while in the paper there seem to be plenty of other benchmarks. The data loaders and the training scripts worked fine. However the conda environment file was faulty which were resolved.
The residual dataset is missing which is understandable, the results on SIXray seem to be reliable.

**Overall Reproducibility Review:**

The notebook is straightforward but it lacks annotation. One of the datasets is public but the other one is not. The number of classes are too few, and given the fact that the performance of nearly all models is low on one of the classes signifies the need for more variability on class list and data gathering areas. I assume that is one of the reasons why the authors extended their work to the proprietary dataset regarded as the residual dataset.
It is recommended to comment the scripts with more details especially since the small parameter size of the proposed architecture is probably the most significant contribution. The only clear access to training is provided by the argument parser and any additional manipulation is not as easy.
The similarity of code structure for benchmark tests and the proposed architecture is very valuable, and authors did a great job there.

**Review Confidence:**

4: You are confident in your assessment, but not absolutely certain. It is unlikely, but not impossible, that you did not understand some parts of the submission or that you are unfamiliar with some pieces of the code or data.

**Review Rating:**

8: Accept, all aspects of this are reproducible with minor effort.

**Review Summary:**

I recommend this paper for acceptance. Although there is a lack of diverse dataset in newer domains but the contribution of the architecture is clear.

**Summary Of Necessary Code And Dataset Supplement:**

The present code includes various SOTA object detection architectures which are regarded as benchmark networks that are NOT domain specific. The authors provided two benchmark networks and the proposed MEOW network.
Nevertheless the proposed architecture produces similar results to what is presented in the paper on the public SIXray dataset. The small size of MEOW architecture is great although the IoU percentage and F-score does not outperform the benchmarks.

**Usability And Ease Of Reproducibility:**

There was something wrong with the env.txt file. The format of text file was not acceptable, at least on the arm64 architecture which was used as the host device. Also, there are too many auxiliary python files present in the repo which is not helpful.
The train, test and data loader notebooks were straightforward.

---

> ### Author Response · Authors · 2023-04-29
> **Thank you very much for your encouraging comments, useful input, and recognition of our efforts.**
>
> We have provided dataloaders for different standard datasets, sorry we did not clarify this aspect earlier, the reason was just that we were particularly focused on establishing the feasibility of such approaches in concealed weapon detection as a critical case study.
> We agree with you, and recognise the need to test our system on various datasets and include standard ones to enhance comparison with other AutoML approaches.
>
> Towards this goal, now we include experiments with 10 different runs for CIFAR-10 since it is the most popular dataset for testing Neural Architecture Search algorithms. Please find the results in Appendix A (Section 7 in the current working version of the manuscript). The manuscript will be fully updated later this weekend, but we wanted to make sure we give you enough time to read our response, and we want to strongly stress that we are updating the manuscript to account for the great points raised by you and other reviewers.
>
> Regarding the comments and the README in the repository, most of them were stripped to comply with the double-blind review process, and the repo has been anonymised. We want to reassure you that the link to the full repository will be published, which has substantially more comments and explanations, sorry for not clarifying this.
>
> Indeed, our main goal is that this proposed technique is widely adopted, and we will provide all the means to facilitate this. We thank you for your input, and we will ensure that the comments and tests in the repository are clear and comprehensive.

---

### Official Review · Reviewer_Sh2H · 2023-04-23

**Potential Impact On The Field Of Automl Rating:** 3
**Technical Quality And Correctness Rating:** 3
**Clarity:** I was confused about some technical d…
**Clarity Rating:** 2
**Actions Required To Increase Overall Recommendation:** Convincing answers to my questions ab…

**Summary Of Contributions:**

The paper proposes an efficient multi-objective NAS method for concealed weapon detection in X-rays. The contributions are twofold. The paper first introduces computationally cheap heuristics to make genetic multi-objective NAS algorithms  for this application efficient and effective. The approach is evaluated on publicly available and private datasets. Secondly the paper proposes an approach to effectively ensemble multiple architectures discovered. The results show that their approach outperforms state-of-the-art methods for concealed weapon detection.

**Overall Review:**

Positives:
The application and the necessity of automl for it is well motivated
The paper studies the effectiveness of automl in a real-world industry application.
Exploiting zero cost proxies for multi-objective NAS is fairly novel to the best of my knowledge

Negatives:
Background and example definitions of “threats and modifications” is nor provided.
It is not clear why optimizing for different proxy scores is formulated as a multi-objective problem. In traditional multi-objective NAS the objectives are often user centric (eg: hardware efficiency, performance quality). Why is optimization of different zero cost proxy scores simultaneously desirable to search for optimal architectures? How were the proxy metrics selected?
Evaluation on a private dataset makes some results presented not reproducible. Could the authors evaluate on more publicly available benchmarks?
The paper does not provide details on the search space for the task. Is it a set of all the architectures/operations from table 1?
For the results table 1,2,3 it would be nice to evaluate across multiple search seeds.
Could the author’s provide an intuition on why the performance gains do not translate to semantic segmentation in table 4?
Presentation of the approach could be improved.

**Potential Impact On The Field Of Automl:**

I find the overall impact on the field of AutoML to be moderate. The application is indeed interesting and different from the typical computer vision applications NAS is applied on. I also find the use of zero-cost proxy scores for multi-objective search novel and interesting.

**Reproducibility (Optional):**

Use of private datasets does reduce the reproducibility of some results presented.

**Review Confidence:**

4: You are confident in your assessment, but not absolutely certain. It is unlikely, but not impossible, that you did not understand some parts of the submission or that you are unfamiliar with some pieces of related work.

**Review Rating:**

4: Weak Reject: For instance, a paper with minor technical flaws, limited impact, and/or weak evaluation.

**Review Summary:**

I find the application quite interesting and well motivated. However I think the paper misses some important technical details and description of terminologies used. This makes reading the paper and deriving connections between different parts difficult.

**Technical Quality And Correctness:**

Overall fine; evaluation on more than one public dataset would be good to avoid drawing conclusions based on noise.

---

> ### Author Response · Authors · 2023-04-29
> **Thank you very much for the insightful comments and useful input. Please find our answers to some of the main points.**
>
> 1. Further evaluation: To boost the generalisability we include a 10-run case study on CIFAR-10 in the updated version. We have chosen CIFAR given its popularity in NAS, which ensures a fair comparison. Please refer to the updated Appendix for the CIFAR-10 results.
>
> 2. Threats and modifications: We have added a brief explanation of the different threats and modifications to the paper as requested. The threat types are similar to the ones of SIXray and the modification types refer to how the particular item has been modified. For example, since the dataset is produced by taking multiple images, let's say the two images are of a scanned baggage and there is a gun in the first and then there is no gun in the second. This will be a threat modification of "gun" and the modification type would be that the gun is "removed". The reason we can't disclose the actual modification types and threats is that this could lead to establishing a link between the industrial project and this research which we are legally not allowed for security reasons, our apologies.
>
> 3. Multiple proxy scores: Thank you very much for your comment and interest in using multiple proxy scores. During the search, we use three objectives: the NASWOT, the SYNFLOW and a complexity score. The last one aims to steer the searching process towards smaller non-over-parameterised networks, rendering the discovery of faster and more compatible architectures, as shown in the results (especially in semantic segmentation). This is now clarified in the revised version.
>
> 4. Proxy scores selection: We briefly outlined (page 5 top) the need for the synergy between the NASWOT score (a data-dependent score) and the SYNFLOW score (a data-agnostic measure). This explanation was admittedly too brief. The motivation behind the choice is that NASWOT was originally designed to evaluate the difference between the linear regions produced by the model on the data at hand. In contrast, SYNFLOW provides a more general overview of the network's capacity by using an input of a vector of ones, regardless of the data and hence they provide "orthogonal", complementary views of the system's performance. We picked the two scores since they are, in our view, the most robust scores in the literature. They consistently score highest when compared to alternatives in studies such as Krishnakumar et al.'s 2022 "NAS-Bench-Suite-Zero" or  Abdelfattah's 2015 "Zero-Cost Proxies For Lightweight NAS". In addition, our suggested approach is compatible with any proxy score or an aggregated proxy of multiple scores at once. As shown in Abdelfattah's 2015 "Zero-Cost Proxies For Lightweight NAS", such a combination of proxy scores can lead to better results since some of them (such as SYNFLOW and NASWOT) carry information about different aspects of the network.
>
> 5. Search space for the task: Thank you for your comment, we are adding these details in the Appendix of the revised version.
>
> 6. Semantic segmentation performance: Thank you for pointing out the need for clarifying further the segmentation results. The reason for these results is the multi-objective nature of the process. The fact that it did not achieve the highest test IOU or F1 score is due to the optimal balance established by the model between these scores and the computational cost.
> Performance gains translated to this problem, but the process suggested that it was optimal to slightly sacrifice accuracy to reduce the model size by several orders of magnitude. Even though these 2 metrics are indeed worse than 2 of the other models, the MEOW model is more than x50 and more than x100 times smaller than the 1st and 2nd ones, respectively, to make it much more computationally efficient and compatible with a plethora of devices and settings.
>
> MEOW is also the only approach of the selection capable of real-time inference, which is crucial for the application it was designed for. Perhaps more important, the analysis of the gap between the validation and test results reveals that the MEOW model is the most robust out of the full selection: note that the difference between the scores is less than 1%, and this is not observed with any of the other models. This property makes it significantly more trustworthy and reliable, hence, superior to the rest of the models. Moreover, if compared to similarly sized models, it does outperform them.
>
> 7. Multiple seeds: All the experiments were run with multiple seeds (5). This was indicated in point (j) in the checklist, this is now clearer in the revised version. In addition, we have also run CIFAR with 10 different seeds. The scope of this paper is to present a proof-of-concept, but, interestingly, it is straightforward to run even more seeds using the provided code in the future.
>
> Thank you again for your comments and we are looking forward to your views and reply. Please find also the revised manuscript.